# Older Adults’ Perceptions of the Usefulness of Technologies for Engaging in Physical Activity: Using Focus Groups to Explore Physical Literacy

**DOI:** 10.3390/ijerph17041144

**Published:** 2020-02-11

**Authors:** Alexandre Monte Campelo, Larry Katz

**Affiliations:** Sport Technology Research Laboratory, Faculty of Kinesiology, University of Calgary, Calgary, AB T2N 1N4, Canada; katz@ucalgary.ca

**Keywords:** physical literacy, technology, active videogame, elderly, focus groups

## Abstract

Insufficient physical activity (PA) levels observed among older adults remain extremely high and pose a danger to developing and maintaining their physical literacy (PL). Each person’s level of PL partly depends on their physical and cognitive skills, confidence level, and degree of motivation to practice PA daily. New technologies, such as exergames and wearable fitness trackers, may enable older adults to increase their PL, stimulating uptake and ongoing PA participation. *Objective*: This focus group study aims to describe older adults’ perceptions of the use of technologies to engage in physical exercise programs. *Methods*: Fifteen participants were randomly selected from a sample of 40 older adults who completed a randomized controlled trial that investigated the benefits of using technology in the context of group-based exercise programs. Separate post-intervention focus groups were performed with an exergaming group, a conventional physical training group, and a no training group (control). Data were mapped onto constructs from the four domains of PL: affective, physical, cognitive, and behavioral. *Results*: Generally, participants expressed positive perceptions about the benefits of using technology to engage in PA. These positive feelings outweighed the costs and the lack of familiarization with technology. Common themes for the three groups emerged from the discussions and included familiarization with technology, using fitness tracker to monitor PA, previous exposure to technology, and interaction with peers, staff members, and relatives. In particular, participants from the exergaming group explored the ideas of training their cognitive skills while using the exergame accessories, exercising in an alternative way, competitive versus cooperative play, changes in sense of humor, skill transferability from game to real environment, progressions of the exercise intensities, and the potential use of exergames for rehabilitation. *Conclusions*: Participants in this study reported positive perceptions about implementing technology into exercise. Emphasizing the benefits of using technology in group-based exercise programs may increase older adults’ PL levels and their future technology adoption. The potential implementation of technology into conventional exercise programs should focus on older adults’ lifelong values, biopsychosocial conditions, and the possibility of reducing age-related risk of injuries and chronic diseases.

## 1. Introduction

According to the United Nations’ report [1], the proportion of people aged 65 and over is growing faster than any other age group worldwide. Following the same trend, the rate of technological development also steadily increased. Advances in technology in healthcare and changes in older adults’ lifestyles led to reductions in mortality and increases in life expectancy [2]. With the progressively changing demographics, technology use amongst the elderly is an important research consideration. 

The aging effects on health behavior, such as a sedentary lifestyle, strongly influence how older adults perceive and experience their surrounding environment [3]. The availability of resources and accessibility to attractive and safe environments can facilitate physical activity (PA) engagement by older adults [4]. However, only 10%–30% of older adults report engaging in regular physical exercise as recommended by validated PA guidelines [5,6]. The prevalence of sedentary lifestyles poses a danger to maintaining physical and cognitive functions, especially when considering older adults’ age-related vulnerabilities.

Identifying key determinants of an active lifestyle and how these determinants can be promoted is central to successful healthy aging [7]. Previous research showed that the combination of physical exercise with cognitively challenging tasks provided by technology-based exercise programs can improve global physical and cognitive functions in healthy and clinical populations of older adults, compared to physical exercise training alone [8,9,10]. Researchers examined possible innovative alternatives for PA promotion, including mobile phone applications [11], wearable and smart home activity sensors [12], and use of active video games or exergames [13,14]. These relatively new technological tools have the potential to be an effective alternative to fulfill sedentary leisure time and enhance older adults’ perceptions of competence, motivation, and confidence to engage in PA [15,16,17]. An appropriate implementation of technology based on a well-established theoretical framework may promote successful technology adoption by older adults [18,19].

As a promising strategy to achieve lifelong participation in PA, physical literacy (PL) could be a meaningful approach to reducing sedentary behaviors and prevent associated chronic diseases [20]. PL is defined as the “motivation, confidence, physical competence, knowledge, and understanding to value and take responsibility for engagement in physical activities for life” [21]. The holistic nature of the PL concept also provides the basis for understanding older adults’ technology adoption and perceptions of its use to engage in PA. 

### Objective

This study aims to examine older adults’ perceptions of the use of wearable and exergame technologies to engage in physical exercise programs.

## 2. Materials and Methods

### 2.1. Study Design

A focus group approach was used to collect perceptions from three groups of participants in a broader research study designed to increase the PL levels of older adults. The research was conducted in a community facility located in Calgary, Canada from August to December 2018. The study was approved by the Conjoint Health Research Ethics Board, University of Calgary, under the reference protocol number REB16-1633. An interpretive description methodological approach was used to guide data collection, analysis, and presentation of results [22]. Data were mapped onto constructs from the four domains of the PL consensus statement: cognitive, affective, behavioral, and physical [23].

### 2.2. Sample and Setting

As part of a larger, randomized controlled trial, 40 older adults were recruited from community and independent-living centers, and randomly allocated to one of the three study groups: exergame training (ET, *n* = 15), conventional training (CT, *n* = 14), and no training (NT, *n* = 11). Inclusion criteria for eligible participants were as follows:Participants aged 65 years and over;Able to ambulate independently without an assistive device;Able to adequately see and hear to interact with the game;Able to read and speak English to follow instructions.

Participants were excluded from the study if they did the following:Failed to clear the cognitive test: Mini-Cog Test, score <3 [24];Presented with an active acute/chronic disease process in the Physical Activity Readiness Questionnaire for Everyone (PAR-Q+) [25];Took part in a PA program within three months before the study started;Were diagnosed with a balance or neurological disorder that seriously affected motor function (e.g., stroke, traumatic brain injury, spinal cord injury, nerve injury, multiple sclerosis, or Parkinsonism).

All participants wore a Fitbit Flex 2™ (Fitbit Inc., San Francisco, United States) for the duration of the study. Participants did not have access to the data from the Fitbit until the end of the study. CT and ET groups downloaded the Fitbit data once a week before training. The participants in the NT group were asked to commute to the community facility once a week to meet with the researchers to download the Fitbit data, but did not have contact with the other groups.

Both the ET and the CT programs were supervised by a physiotherapist and an assistant. Participants in the ET group used the Nintendo Wii-U and its accessories, such as the Wii Remote and Wii Balance Board to perform the exercise program with the Wii Fit-U sub-games. The CT program consisted of exercises done seated on or standing behind a chair and laying down on an exercise mat. Participants from both exercise groups were asked to attend every session conducted three times a week for six weeks. ET and CT participants were divided into three groups each (six groups total) with approximately five participants in each group. In the facility, two similar exercise rooms were used with three time periods, two in the morning and one in the afternoon, so that training was matched for time and group size. The participants in the NT group did not engage in any exercise training and were advised to maintain their usual daily routine.

Within one week of the end of the intervention period, six participants from each group were randomly selected and invited to participate in focus group interview sessions. Separate post-intervention focus groups were performed with each study group. Participants signed a consent form agreeing to participate and allowing the moderators to audiotape the conversations.

The principal investigators created an open-ended, semi-structured interview guide. The topic guide for all the three focus group interviews included a guiding question and prompts in the following areas: General impressions about the study: “Please describe your overall experience throughout this program.”Using technology for PA engagement: “Do you think that technology can help you to be more active?”Physical literacy perception: “After describing Whitehead’s definition of PL, do you think that your PL level changed throughout the study (last six weeks)?”Future technology adoption: “Would you be interested in continuing to use technology in your physical activities in the future?”

The focus group sessions were led by a trained moderator and a note-taking assistant, who did not participate in the exercise programs. Refreshments were served to compensate the participants for their time.

### 2.3. Data Analysis

Immediately after each focus group, the moderator and assistant discussed the group dynamics, noting common themes and unexpected items. The audiotapes were transcribed verbatim and the transcriptions were checked for accuracy against original recordings. All participant identifiers were replaced by a study code to protect their confidentiality. Transcripts and research field notes were the data products used for analysis.

Interpretive description methodology [22] and analytic procedures of thematic analysis [26] were applied to the three focus group transcripts and field notes. The process of codification was mixed: deductive and inductive. It was deductive as the main themes were coded, categorized, and mapped according to the constructs of PL. However, due to the genuine interest in the raw data to reveal possible new main themes, the inductive approach was also applied. Subthemes were created to support the ideas that emerged from the main themes. The main themes and their subthemes uncovered in the analysis were uploaded to NVivo12 software (QSR International, Doncaster, Australia). The software allowed the researchers to organize the data and tabulate participants’ references by theme. The emerged themes and subthemes are presented in the results section through the interpretation of participants’ own descriptions of their experiences.

## 3. Results

Focus group sessions lasted an average of 38 min (ET: 59 min, CT: 31 min, NT: 24 min). Of the 18 invited, 15 participants (ET, *n* = 6; CT, *n* = 4; NT, *n* = 5) attended the post-intervention focus group interviews. Three participants (CT, *n* = 2; NT, *n* = 1) did not show up for their focus group session. Two gave no reason and one had the wrong day in his calendar.

Ten focus group participants (ET, *n* = 3; CT, *n* = 3; NT, *n* = 4) identified as female (66.6%), representing 37% of female participants in the main study (*n* = 27). The average age of the focus group participants was 73.53 years (ranging from 65 to 85 years), compared to 72.6 years (ranging from 65 to 95 years) from the main study. Participants reported being independent community-dwelling older adults and two of them (CT, *n* = 1; NT, *n* = 1) were residents in an independent-living facility.

### 3.1. Themes: Physical Literacy Domains

Focus groups provided a rich dataset, illuminating older adults’ perceptions of the use of technologies to engage in the physical exercise programs. As recommended by Smithson [27], individual opinions were not treated as belonging to individuals within the group, nor as held by the whole group, but as constructed discourses that emerged in social and dynamic conversations. In addition to the well-established PL domains, the inductive approach yielded the social/economic domain as a main theme, given its importance as a determinant of health behavior [28]. Thirty-six subthemes were linked among the overarching umbrella themes, as presented in Figure 1.

#### 3.1.1. Cognitive Domain

As depicted in Figure 1, the cognitive domain was influenced by eight subthemes that centered on participants’ cognitive capacities to understand and interact with the technologies used to engage in PA. The subthemes shared by the three groups included familiarization with technology and using a Fitbit to monitor PA. Despite being a shared subtheme between the three groups, participants demonstrated different levels of familiarization with technology. For example, P3 (male, 66) from the ET group reported the following: “This study introduced me to technologies that I’ve never even heard or seen before.” P2 (ET, female, 72) presented her frustrations when previously trying to engage in PA using an exergame device: “I was involved with technology since I was young… So, I thought I knew all these things. Well, I bought a Sony [active videogame] because I wanted to get fit and it has a fitness program. But I didn’t know how to use it. I used it for about a day and gave up.” In the meantime, P11 (NT, female, 73) demonstrated more familiarity with technology: “I already had a Fitbit. I was afraid that it could interfere with yours, but it didn’t at all.”

The subtheme named knowledge and understanding of PA benefits was shared between participants from the CT and NT groups. P7 (CT, male, 73) mentioned the following: “When I was young, there was nobody who taught us: ‘oh, you should be exercising three times a week.’ Just hearing about your study with technology already makes a big difference.” P8 (CT, female, 70) commented on her perceptions in regards to the feedback provided by the Fitbit: “It showed me what I was doing and, you know, what [parameters] I need to improve a little bit more.” Meanwhile, the participants from the NT group expressed their curiosity about the exergame training: “Do you have different expectations according to the age of the participant?” (P12, NT, female, 72). 

Participants from the ET group further explored the ideas of training their cognitive skills while interacting with the exergames. For example, P4 (ET, female, 66) described her experiences while playing a dual-task game: “Several times I was like, ‘okay, I’ve got to remember to always look on that top one [bar].’ Because if I was only working on the one below, I learned that my score went down. It was probably good for my brain.”

#### 3.1.2. Affective Domain

The affective domain was reflected by the participants’ perceptions of their confidence, motivation, and enjoyment in relation to their technology adoption to engage in PA. For example, P10 (CT, female, 73) stated the following: “I have got a Fitbit and that got me motivated because I could see my [number of] steps and it challenged me a lot… I think it [Fitbit feedback] really makes you feel better about yourself.” P6 (ET, male, 65) shared his perceptions after engaging in the exergaming program: “I think it was great. I am disappointed it’s coming to an end, but it really got me motivated again… So, I’m really glad for the luck that I got in the Wii group because it was way more fun than standing over there doing exercises, you know, seriously.” 

Participants who engaged in the exergame training discussed the role of competitive versus cooperative play on their experiences. For example, P3 (ET, male, 66) mentioned the following: “There was some level of competition between us. It was fun though. I also had fun when we had a common goal and it was that tunnel”, referring to the game in which the participants were asked to statically walk on the balance board, lifting their feet in order to mimic the motions of pedaling a bicycle. P4 (ET, female, 66) replied the following: “Well, it was a supportive competitiveness.” The discussion continued with P2 (ET, female, 72) expressing her point of view: “It’s a bit of competitiveness with yourself. You know, when I did this the last time, I was like, ‘oh, I got to another level.’ I felt good.” 

In addition, participants in the ET group were grateful for being able to participate in the study, identified a change in their sense of humor, and wished that the participants from the other groups could have had the same experience: “We [participants from ET group] sort of ended up melding in our sense of humor with a bit of competitiveness sometimes. The control group is probably disappointed.” (P4, ET, female, 66).

#### 3.1.3. Behavioral Domain

The analysis of the behavioral domain showed that participants from the three groups were able to compare their previous exposure to technology with the current experience. P11 (NT, female, 73) mentioned that she had a fitness tracker device before starting the study: “I had a Fitbit before. So, I could set a goal of two hundred steps every hour between 7:00 a.m. and 7:00 p.m. If I didn’t complete them, it would giggle and tell me I’ve got to get up and exercise. I couldn’t do the same [setting] with yours [Fitbit].” In addition, P7 (CT, male, 73) mentioned the following: “I used to be active a number of years ago and the last few years I just haven’t been. So, it was a great chance to participate, get more familiar with the Fitbit, and sort of moving forward from being sedentary.”

Participants from both exercise training groups presented a higher sense of commitment to wear and charge their Fitbit while attending the sessions three times a week, compared to NT group. For example, P5 (ET, female, 75) mentioned the following: “I never took it off, except to recharge it.” P8 (CT, female, 70) also stated the following: “It [The Fitbit] kept me engaged in the study. Like, I felt the responsibility to recharge it and come to the sessions. If you volunteer for the study, you feel the responsibility to do it. It’s like you’ve paid for a device or for a fitness class. So, you want to get your money’s worth.” One of her peers (P9, CT, female, 72) agreed: “Yes, when I paid for a trainer, I found lots of reasons not to go and I still ended up paying for it. I’ve never paid for this [study] and I never missed a single session.” However, the majority of the participants from NT group agreed that they forgot to recharge the Fitbit often. P13 (NT, male, 73) tried to justify why he forgot to recharge his device: “Half of the time I forgot the thing [Fitbit] was there. I think it is because we didn’t have access to our accounts, and we had to come only once a week. There was nobody to remind us [about recharging the device]. So, as a result, we didn’t charge it when we should have done so.”

ET participants, in particular, expressed their perceptions of trying a new way of exercising. P1 (ET, male, 73) reported the following: “I like the variety of them [Wii Fit-U sub-games]. I could tell my body was using different muscles. When you’re not doing anything, you think ‘they are all right’. Then, you use them and you think ‘oh, that’s a new muscle’.” P5 (ET, female, 75) hypothesized the following: “I loved the Wii. If I had one, I could just sit it in front of the T.V. and sometimes I would flip the switch to exercise.”

#### 3.1.4. Social/Economic Domain

As for the social/economic domain, participants from the three groups shared similar perceptions about the interaction with their peers, staff members, and relatives during the study. Participants from the intervention groups enjoyed exercising with their peers. However, participants from the ET group perceived that their interactions were more related to socializing while playing the exergames. For example, P2 (ET, female, 72) stated the following: “I never missed one of these sessions at all. A lot of it was because of the people, to be honest with you.” P1 (ET, male, 73) engaged in the conversation and continued: “It was more than just the exercise that we were doing. It was the people that we were working with. The staff members were just incredibly great to have in the room, because they were always supportive and they were always there for us.” Participants from the ET group spoke about their attitudes towards playing exergames with their grandchildren. P4 (ET, female, 66) mentioned the following: “One [participant] of our group is going to buy one [Nintendo Wii-U]. Her nephew has one and he suggested to her to buy it. So, it’s another way to interact with our grandchildren.” Participants from the CT and NT groups demonstrated their gratitude for the support given by the staff members in regard to issues with their fitness trackers. They also revealed that they sought help from their relatives with the placement of the device.

The cost to purchase the equipment used in the study was a shared subtheme that emerged from the conversations between participants that engaged in both exercise programs. Participants focused on their perceptions of the cost–benefit ratio. P1 (ET, male, 73) summarized as follows: “I might consider even buying one [Nintendo Wii-U]. It doesn’t matter its cost.” 

The subtheme named willingness to try new technologies was brought up by the participants from the NT group. Based on the comments, participants who did not engage in any of the exercise training seemed willing to try the exergames. Moreover, they appreciated the caring context in which the study was offered.

#### 3.1.5. Physical Domain

In the physical domain, participants from the CT group expressed their perceptions about using technology to monitor their physical competencies, such as their heart rate and number of steps. P8 (CT, female, 70) who owned her own Fitbit, commented on how the device was helpful to her: “Because I’ve got a really low heart rate, I’ve got to move more. So, the Fitbit really helps me to monitor these things, like my BPMs [beats per minute], number of steps, and so on.” 

Participants from the ET group presented different perceptions about their experiences using the Wii Balance Board. For example, P3 (ET, male, 66) and P4 (ET, female, 66) shared their frustrations while using the exergame accessory: “Last Friday, we had two boards that were almost dead on battery. They just were not working properly. It was just really a frustrating day for some of us.” (P3); “I felt bad because I got to a new level on that game on Wednesday, but when I tried it again on Friday, I couldn’t get closer [to the same level]. That’s why I was frustrated.” (P4). In addition, P1 (ET, male, 73) stated the following: “The board’s malfunction was the only complaint, and that’s not your guys’ fault. You don’t build those boards, and I don’t think they were that great.”

Participants from the ET group discussed their previous experiences engaging in conventional exercise training as opposed to the exergame training. P1 (ET, male, 73) shared his perspective: “When I was paying a trainer to do the job, part of the benefit was because I was being taught a proper form. [*Trainer’s name*] was always on me to do it properly. That’s one thing you don’t get with the Wii. I mean, if [*staff members’ names*] weren’t here correcting our forms, I wouldn’t correct it myself.”

The idea of skill transferability from the game to the real environment was another subtheme that emerged from discussions between the participants of the ET group. P4 (ET, female, 66) shared her perspective about the meaning of the scores provided by the exergames: “I had to keep reminding myself that getting a higher score didn’t mean I was getting fitter. It meant I was getting better at the game. So, I had to really remind myself of it.” One of her peers (P5, ET, female, 75) disagreed and stated: “I think you could be getting fitter.” P4 (ET, female, 76) replied, “I could be getting fitter, but that’s not really the feedback from the game. The feedback was that I was getting better at the game. And that may be part of its seduction. It lets you think you’re getting fitter because you’re getting better at the game. So, I’m not saying that’s a bad thing, but I had to be realistic of the fact that this really wasn’t an indication that I was getting fitter.” P2 (ET, female, 72) was more specific and described her experience when balancing on the Wii Balance Board: “I thought I was very well-balanced because I have been skiing for 59 years. I’m still skiing, and I don’t fall or anything. So, when I tried to stand on one foot, it was just like, ‘I don’t ski on one foot’. So, for me it’s been a bit challenging.”

Participants from the ET group were able to perceive different skill levels according to the games’ task and the simulated environments where they performed their training. “I could do the super hula hoop. For me, that was a piece of cake. But, for other people in the group, you know, I just drove a couple of them crazy. They just couldn’t do it. Then, I couldn’t get the ski or the bird games. But they would get them. So, yeah, I could be really good at one exercise but not at another ones.”

Participants from the ET group also discussed their perceptions about the exercise intensity, its progression during the six-week programs, and its potential use for their physical rehabilitation. P2 (ET, female, 72) described her perceptions of the exercise intensity: “The activities were short, but they were intense.” P4 (ET, female, 76) elaborated her thoughts about the exercise intensity relating to the progression of the difficulty in one of the games: “It’s not as hard as the workout that I do in the gym, in terms of heart and respiration rate. But, near the end, I couldn’t repeat [the game] three times. If I did three, I felt that I was working out somewhat near maximum [exertion].” Another participant (P6, ET, male, 65), who had a knee replacement, mentioned his perspective of the potential use of the exergames for his physical rehabilitation: “I had to get my knees replaced two years ago. So, I’m not supposed to run. But on that activity [running on the spot], I’m not really running. I’m not putting a lot of pressure on my knees and I felt great. I will ask my physio to include these [activities] in my therapy.”

## 4. Discussion

This qualitative study explored older adults’ perceptions of using technologies, such as exergames and fitness trackers, to engage in physical exercise programs. The focus groups’ analysis was primarily guided by an interpretive description of older adults’ perceptions under the PL theoretical framework. This methodological approach was initially developed for clinical contexts; however, in the study described in this article, the participants were physically independent and did not present any functional impairments. Here, each main theme is discussed from the standpoint of how technology may be utilized to promote lifelong PA engagement and its potential use in the prevention and rehabilitation of a variety of diseases.

From a holistic perspective, Whitehead [29] proposed that PL encompasses more than a motor action. A physically literate individual should not only be able to “do” a PA, but also be able to “read” the environment and respond appropriately. The “reader” relies on a range of cognitive skills as it resonates with previous knowledge and experience [29]. The technological tools used in the current study were able to quantify and provide feedback on participants’ PA performance, which assisted them with “reading” their surrounding environment.

Individuals exchange knowledge, experiences, values, and beliefs in constant adaptation to the environment in which they are living. Many older adults are not comfortable with technologies; therefore, adoption may be a challenge [19,30]. Moreover, only 36% of older adults are aware of the PA recommendations [31]. Providing knowledge and understanding about the importance of PA for a healthy lifestyle, the need to be physically literate, and the role of new technologies in facilitating PA and PL can reduce the barriers to PA engagement.

Familiarization with new technologies is fundamental and influences the success of implementation to engage in PA programs. Research indicated that, despite becoming increasingly familiar with technology, older adults generally have different abilities and competencies compared to their younger counterparts [32,33]. The relatively recent development and widespread use of wearables and exergames may account for this disparity in abilities and comfort. Findings from the three focus groups demonstrated that the participants presented different levels of familiarization with the technologies used in the study. However, Loew et al. [34] stated that the level of knowledge does not necessarily translate into long-term adherence in physical activity engagement.

Researchers investigated the factors that employ a strong influence in older adults’ attitudes toward and adherence to the use of technology to engage in PA. Matz-Costa et al. [35] demonstrated that the affective characteristics that are developed across the lifespan, such as the perceived feelings of enjoyment and satisfaction from previous experiences, the control over exercise, personality, and psychological health, better predict higher levels of PA adherence. According to the self-determination theory supporters [36], the intrinsic and extrinsic motivations have a significant impact on individuals’ attitudes toward PA engagement. Intrinsic motivation involves participating in an activity for the enjoyment and satisfaction inherent in engaging in continued behavior. In contrast, external motivation is a state in which individuals’ behavior is controlled by specific external factors [36]. Several studies indicate that individuals who exhibited self-determined types of regulation show more persistence in PA engagement [37], exercise adherence [38], PA intentions [39], and enjoyment of physical exercise [40]. Participants in the current study reported that wearing the Fitbit and playing exergames were enjoyable experiences and expressed their wishes to continue the exercise programs. In particular, participants from both exercise groups identified intrinsic and extrinsic motivators to engage in PA. Intrinsic motivators included the wish to get fitter and reduce functional impairments, intentions to learn how to use the Fitbit and how to play the games, paying attention in certain parts of the game, sense of humor, and self-confidence. Extrinsic motivators included the cost and portability of the equipment, access and feedback from Fitbit parameters, sense of improvement, interaction with family, peers, and staff members, competition/cooperation between peers, variability of activities, game scores, and perception of exercise intensity. In contrast, participants also identified external factors that influenced their lack of motivation or commitment. For example, participants in the NT group justified the reasons for forgetting to recharge their Fitbit with the fact that they were not engaged in an exercise program. ET group participants expressed their frustration with the technical issues, such as machines not working properly. These findings suggest that structured exposure to new technologies and experiencing the health benefits may be helpful in motivating initial involvement and future adherence to a regular PA program.

Gothe [41] showed that self-efficacy and outcome expectations are the most consistent affective mediators to change exercise behavior in older adults. The development of self-efficacy is related to individuals’ self-confidence and their personal choices. As people age, their self-efficacy represents their belief in the capacity to trust and exert control over their lives. In contrast, a lack of confidence in physical abilities and fear of injuries are strong predictors of PA avoidance [42]. Participants from the current study expressed that they felt good about themselves after engaging in the exercise programs using the technologies and none of them had falls or injuries during the exercise sessions. Participants who did not engage in any of the exercise training demonstrated their willingness to try the exergames. Despite the positive perceptions toward technology in the current study, health professionals must assess older adults’ self-confidence to adopt technologies. Providing cautious fall prevention procedures, and emphasizing perceived confidence and benefits to exercise may encourage older adults to engage in effective PA and accommodate their technology skills. 

Whitehead [29] considered psychological factors such as the cognitive and affective domains to be the precursors to the physical and behavioral domains. By recognizing the role of cognitive and affective capabilities, Whitehead [29] acknowledged the role of interacting with the social and physical surroundings. Boyle et al. [43] further stated that the affective values arise from the interaction with others and the response to situations in which the individuals find themselves. Recently, Warner and French [44] demonstrated that older adults with high self-efficacy levels are more likely to interact socially, which in turn can facilitate the development of their physical competence within a wide range of environments. It was suggested that older adults with high self-efficacy are less socially anxious and more likely to use technology in general [45]. Thus, adopting technologies may help older adults remain active and socially involved.

The social interaction provided by group-based exercise programs proved beneficial for older adults to overcome psychosocial conditions, such as loneliness, depression, and low self-esteem [46]. Moreover, studies showed that there is a strong correlation between the use of exergames and positive mood among older adults living independently [47]. The participants from the current study demonstrated that they enjoy interactions with their peers, especially the ones allocated in the ET group who perceived the feelings of competition, cooperation, gratitude, concerns for others, and a change in their sense of humor. Participants also recognized the positive contribution of staff members for their support and identified the feasibility of playing the exergames with their family members, such as their grandchildren. These findings provide support for the implementation of technologies as reliable tools to increase general social interaction.

Theories on the relationship between the accessibility to technology, its regular adoption, and possible cost savings in healthcare increased substantially [48]. The theory of diffusion of innovations [49] holds that older adults are less likely to adopt new technologies unless they view the clear benefits of using them. The technology acceptance model also suggests that older adults may be willing to adopt new technological devices when the usefulness and usability outweigh its costs [50]. By increasing access and reducing costs, technology has the potential to trigger behavioral changes and increase perceived physical competence.

The relatively low cost to purchase equipment and a sense of commitment to the study were shared subthemes between participants who engaged in the exercise programs. These findings are consistent with previous studies suggesting that the perception of potential benefits was more indicative of technology acceptance than a perception of cost [45].

Adopting technology and maintaining regular PA engagement by older adults might be the most important challenge. While older adults may have ceased regular PA engagement or believe they would not be able to partake in PA, participation in such technology-based experiences can give them a sense of involvement in physical exercise and “keeping up” with modern times. Older adults’ behaviors tend to remain in patterns that were perceived as constructive in the past [51]. According to Whitehead [29] (pp. 161), “where negative views are repeatedly and forcefully expressed either about the value of physical activity or the embodied competences with which a person is endowed, it may be difficult or impossible for participation to continue.” The participants from this study were able to compare their previous exposure to PA and technology with the current experiences. Despite the positive attitude and willingness to adopt technology in the future, participants also demonstrated their frustrations with previous and current experiences. Therefore, mitigating negative experiences by providing technology maintenance and formal training may introduce and keep technology into older adults’ lives in an accessible way.

As relatively new technological tools, exergames are increasingly being used as an alternative to conventional rehabilitation-based exercises to improve daily PA levels and increase physical fitness in older adults [52]. Research showed that older adults exposed to exergame training are able to improve a variety of physical functions, such as balance control, cardiorespiratory fitness, and gait speed, among others [53,54]. Exergame programs also provide light-intensity exercises and elicit significantly greater energy expenditure when compared with a resting state [55]. The transferability of physical performance from simulated to the real environments needs further investigation; however, evidence supports that older adults perceive lower levels of exertion when exergaming compared to conventional training [56]. In the current study, participants from the ET group had different motor skill levels. However, they were able to adapt to the intensity and progressive difficulty level of the activities along the six weeks. They also compared their experiences interacting with virtual environments, which simulated real activities and engaged in activities which they previously perceived to be impossible because of their health conditions. These findings support the relevance of using technology-based interventions for PA promotion, leisure-time entertainment, and physical rehabilitation of older adults by motivating them to participate in meaningful activities and maintaining their motor and psychosocial skills.

### Considerations and Limitations

Although the participants from the current study enjoyed their PA experiences with new technologies and emphasized the likelihood of using them in the future, the results should be interpreted with caution, given the possible response bias of participants in the focus group settings and the type of technology used in the study. Firstly, the sample was small and predominantly female (66.6%). As such, the interpretation of overall responses may not apply to males. Amagasa et al. [57] demonstrated that men tend to decrease leisure-time PA participation as they age, compared to their women peers. Despite that, a gender discrepancy in terms of technology adoption may reflect societal tendencies. Wilson [58] and Seifert et al. [59] found that males are more likely to use or own technological equipment compared to females. Therefore, it is likely that fewer males were considered novices, which could influence the interpretation of their familiarization with technology and future adoption.

Secondly, the sample included in the current study was considered healthy, independent older adults (age average of 73.53 years). It is important to consider that older adults constitute a diverse group due to their age-related functional declines. There is a possibility that the participants who signed up for the study were already physically active, presenting high motivation and confidence levels prior to engaging in the exercise programs. Thus, considering a larger sample size and further stratification of older adults’ characteristics, such as marriage and employment status, lifestyle, ethnicity, educational achievements, political affiliation, and physical–psychosocial impairments, could provide new insights into the research question, and potentially mitigate gender and age bias.

Thirdly, a broad range of technologies that aide in PA engagement were developed, including commercial and customized types of fitness trackers and exergames. It is important to consider that developers of commercial devices usually do not target older adults as their main costumers. The technologies included in this study were selected based on their pragmatism to answer the research question and on their commercial accessibility. Furthermore, due to the relatively short duration of the exercise programs (six weeks), the participants might have had more positive perceptions because of the novelty of the activity. 

Including the inductive approach to data analysis aided the exploration of the raw data to reveal the social/economic domain as a main theme. Despite the common application to younger populations, the constructs of PL recently emerged as a promising strategy to increase lifelong PA participation for older adults [20]. Current evidence supports the use of the PL concept with older adults focusing on the following challenges:Increasing their health-related quality of life;Achieving recommended PA amount;Continuing their participation in social, economic, and cultural activities [20,28,60].

## 5. Conclusions

In general, participants presented positive perceptions about implementing technology into their exercise programs. Participants expressed concerns regarding their lack of familiarization with new technologies and socioeconomic factors, such as the accessibility and the cost of the devices. Intrinsic and extrinsic motivators were identified as stimulating uptake and ongoing PA participation.

As mentioned, there are a number of limitations related to sample characteristics and the novelty of the experience. Nevertheless, the findings from this study could help inform technology developers about older adults’ preferences in order to design programs and equipment for this specific population. In addition, practitioners may use the results to consider possible adaptations to commercially available technologies, thereby increasing the potential for older adults’ acceptance and regular adoption.

Mitigating negative experiences by providing formal training and accessibility to technology may introduce and integrate these opportunities into the lives of older adults. Emphasizing the perceived benefits of using technology in group-based exercise programs may be helpful in developing and maintaining older adults’ PL levels, and their likelihood of adopting technology to assist with exercise activities. The potential implementation of technology in conventional exercise programs should focus on older adults’ lifelong values, their biopsychosocial conditions, and the possibilities of reducing age-related risk of injuries and chronic diseases.

## Figures and Tables

**Figure 1 ijerph-17-01144-f001:**
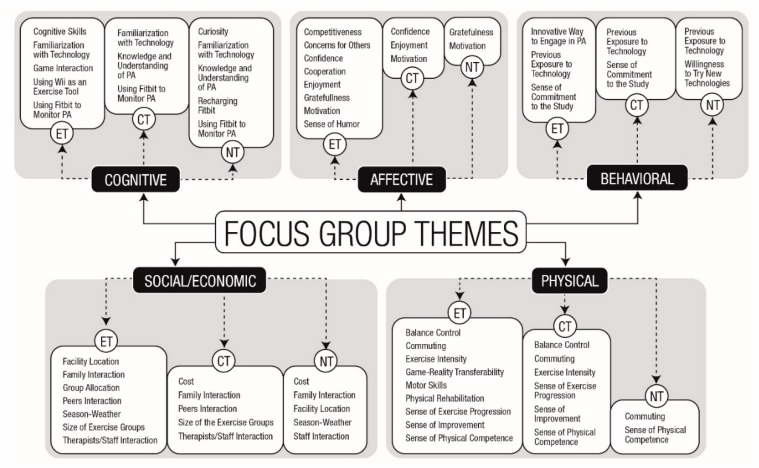
Themes and subthemes that emerged from the discussions of each focus group. Abbreviations: PA—physical activity; ET—exergame training; CT—conventional training; NT—no training.

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
