# Peer review of "Older Adults’ Perceptions of the Usefulness of Technologies for Engaging in Physical Activity: Using Focus Groups to Explore Physical Literacy"

_ijerph, 2020, doi:10.3390/ijerph17041144_

Round 1
Reviewer 1 Report
It is an original study that can be improved. The sample is scarce, but we understand that for such a study it is difficult to find more subjects.
It is recommended to include a section of Objectives, clearly stated, both in the summary and in the article.
It would be important to cite works by Spanish authors who read and write in these magazines. We recommend the Journal of sport and health research that contains 7 articles on this topic.

Reviewer 2 Report
This manuscript is important to know not only researchers but clinicians.
However, this manuscript need to revise several points.
1) Introduction
What is the hypothesis for this study? Please provide details. 2) Methods Since the sample is too small, so, please describe the sample size setting. Based on that, please consider the statistical method again. 3) Results Statistical techniques can cause differences in results.Therefore, please reconsider the addition of data, statistical methods, and the results. 4) Discussion In my opinion, the results can be different.
Therefore, reconsider the consideration based on the results. 5) Other Figure 1 is not easy to understand. Please write clearly again.
Reviewer 3 Report
ijerph-711870
Older Adults’ Perceptions of the Usefulness of Technology for Engaging in Physical Activity: Using Focus Groups to Explore Physical Literacy
Thank you for the opportunity to review your manuscript.
From reading your paper I understood that this was part of a larger study and that you found from your part of the study that older adults were empowered and motivated to use technology with their PA. Your mapping of your data against the Physical Literacy domains seems to suggest that technology use contributes to improved PL.
Comments
I felt that your introduction and statement of the problem were good - nice and concise. Your methods and method of analysis are again succinct and generally well described.
I felt that your results were overly long and wondered whether these could benefit from being edited down and more judicious use of participant quotes to emphasise key points. This section was not helped by some awkward writing and lengthy phrases. Similarly I found the discussion long and had difficulty linking points back to your results. I question whether results and discussion may be better combined rather than as separate sections.
Limitations section was helpful and the conclusions were appropriately guarded and clear.
Overall I thought the science was good but the presentation detracts from your work.
More specific comments are
there are numerous grammatical errors and awkward phrases that need careful proofing and editing. e.g. phrases like ‘highly essential’ and ‘considerably different’ don’t add a lot of information e.g. l 362 As many older adults have not been raised around the new technologies to engaging in purposeful PA? its adoption is not a valued priority for them (doesn’t make sense) where were the exercise sessions conducted? at the retirement facility? the NT group having to commute suggests an external facility? were sessions supervised? were the ET and CT groups exercising in the same place at the same time?(one comment suggests yes) does Table 1 add anything? could that information be text (or could the table replace the text) - seems you have repetition. the brief (6-week) programme was not considered as a limitation - novelty factor?
Best wishes
Round 2
Reviewer 2 Report
Thanks for this revision.
This manuscript will be published after spell check.